# Contact lens procurement and usage habits among adults in Sudan

**Yazan Gammoh**[1]*, **Mustafa Abdu**[2,3]

**1** Faculty of Allied Medical Sciences, Department of Optometry Science, Al-Ahliyya Amman University, Amman, Jordan, **2** Faculty of Optometry and Visual Sciences, Department of Contact Lenses, Al-Neelain University, Khartoum, Sudan, **3** Faculty of Applied Medical Sciences, Department of Optometry, University of Jeddah, Jeddah, KSA

* gammohyazan@yahoo.com

## Abstract

### Objective

The study aimed to assess contact lens (CL) wear and care habits among adults in Sudan.

### Design

An observational, non-interventional, multi-center, cross-sectional study was conducted in the CL departments of all the eye hospitals and centers, and optical centers in the Khartoum State, Sudan.

### Participants

Established CL wearers residing in Khartoum State, Sudan.

### Main outcomes and measures

CL wear profile, CL usage habits, hand and CL hygiene habits were assessed using an interviewer-administered questionnaire.

### Results

The average age of the 442 participants was 24.57 (± 4.87) years. Women comprised 92% of the sample. Smoking was reported by 15.8% of the sample. 81.9% wore soft spherical CL, with 43.4% of the participants were prescribed CL on a yearly replacement schedule. Only 68.3% were prescribed CL by an eye care practitioner. Multipurpose solutions were used by 78.8% of the sample to clean CL, and by 52% to clean the lens case. Compliance rate for CL wear and care among participants was 81.1%. None of the participants reported sharing CL or CL case and rinsing the case with tap water. High compliance level was observed for overnight wear; swimming and showering with CL; handwashing before CL insertion; and cleaning of CL. Moderate compliance rates were recorded for cleaning and replacing lens case, and topping up or sharing solution. Low compliance was noted for attending after care visits.

**Data Availability Statement:** All relevant data are within the manuscript and its Supporting Information files.

**Funding:** The author(s) received no specific funding for this work.

**Competing interests:** The authors have declared that no competing interests exist.

## Conclusions and relevance

CL wearers in Sudan exhibit high to moderate levels of compliance to most contact lens wear and care aspects, except for attending aftercare visits. CL practitioners in Sudan are encouraged to prescribe CL appropriate to the lifestyle and economic situation of patients and actively recommend care products. In addition, practitioners need to follow up with patients to attend aftercare appointments, where hygienic wear and care habits should be emphasized.

## Introduction

Contact lenses (CLs) are used by millions of people worldwide, mainly to correct refractive error, and for therapeutic or cosmetic reasons, with well safety records when used appropriately as recommended by an eye care practitioner [1–3]. Soft CLs are the most common type of CL prescribed or fitted globally, while gas-permeable (GP) CLs represent a small proportion of the type of lenses fitted [4]. Despite the fact that hydrogel soft CLs do not represent more than a tenth of CLs fitted in countries with an established CL market [4], they remain a popular choice by CL wearers in developing markets [5,6]. In addition, cosmetic contact lenses (CCL) are commonly enjoyed by young adult women in Middle Eastern and Asian countries as a common method of cosmetic enhancement [2,3,7,8]. It is of interest to understand the profile of CL wear and care in developing markets to allow eyecare practitioners in these countries to comprehend the demographics of CL wearers, and their attitude towards CL purchase and use [6].

CLs are considered one of the most widely used medical devices globally and they require a prescription from an eyecare medical practitioner in many developed countries [9–11]. Nevertheless, CLs in many countries are sold without a prescription and obtained from various retailers such as optical stores, pharmacies, beauty salons, and over the internet [7,8,12]. Furthermore, it has been noted that eye care practitioners in developing countries do not actively prescribe CLs in comparison to spectacles, neither do they routinely recommend the appropriate CL care products. Additionally, the patient education on appropriate use of CLs is lacking [6,12–14]. Gauging practitioners' readiness to advise and educate patients is key to understand the level of compliance to CL wear and care [15].

Historically, CL wearers, especially young adults, exhibited noncompliance to at least one aspect of CL wear and care [16]. Commonly cited areas of non-compliance include infrequent cleaning and discarding of the lens case, improper use of lens care products, use of tap water while wearing or cleaning CLs, and not adhering to a proper handwashing routine [17]. While certain personal habits such as smoking and hygiene can be modified to enhance compliance to CL wear and care, age, gender, and level of education are risk factors for non-compliance that are non-modifiable [15,18,19]. Of importance to monitor and evaluate compliance is attendance to aftercare visits [20], where adherence to scheduled visits have been observed to be low, especially among CCL wearers [8,12].

Despite the advances in CL materials and the simplicity of care products, non-compliance to CL wear and care remains prevalent amongst wearers in many regions [9,18]. Many studies have investigated the level of compliance to CL wear and care, focusing on specific groups of CL wearers such as daily disposable (DD) CL wearers [21], females [7], or students [12,22]. Nevertheless, few data on CL compliance have been derived from studies that are more

representative of the country's population [8,18,23]. Differences in the methodology used to assess and measure compliance hinder the ability to reliably compare the compliance outcomes of various studies [17,18,24,25].

There is scarcity of information on the profile of CL wearers and the levels of compliance to CL wear and care from developing markets such as countries in Africa [6,26]. Moreover, there are no data available, to date, on the profile of CL wearers in Sudan including CL procurement and usage habits. Determining the CL wearers profile in addition to the compliance level of CL wear and care behaviors could prove useful for eyecare practitioners to evaluate their strategies in the provision and care of CL to the Sudanese population. The aim of the study was to understand the purchase, wearing patterns, and care of CLs. In addition, we also evaluated the levels of compliance to CL wearing guidelines amongst a group of adult CL wearers in Sudan.

## Materials and methods

### Study design and population

An observational, non-interventional, multi-center, cross-sectional study was conducted between January 2019 and February 2020 in the CL departments of all the eye hospitals and centers in the Khartoum State, Sudan. In addition, participants were recruited from optical stores in the area. The state of Khartoum was chosen as it has a diverse population from other states who have settled within the state. In addition, most of the CL departments in the country are found in Khartoum, thus providing easier access to CL wearers.

Adult CL wearers 18 years of age and above, attending the CL department/clinic were invited to participate in the study by research assistants who are licensed optometrists trained by the investigators to interview the participants using a structured questionnaire (S1 Table). Patients who have been wearing CLs for at least a month were included in the study [27]. However, patients with active ocular pathology or who were not able to understand the instructions prior to the administration of the questionnaire were excluded from the study. All patients attending the clinics during the data collection period were invited to participate in the study. To further minimize selection bias, and to ensure full participation of patients, volunteer optometrists administered the questionnaire rather than the patients contact lens practitioners. Clinics in Khartoum operate from 9 am to 7 pm, however, volunteer optometrists approached all the patients attending the contact lens clinics between 10 am and 4 pm to allow for safe and secure transportation of volunteers. The same convenient sampling hours were also adopted for optical stores which usually operate until 9 pm. Approximately 20% of the optical stores in Khartoum were not surveyed due to difficulty in organizing secure transportation for the volunteers. To the best of our knowledge, there are no published data on the prevalence of refractive error nor the number of contact lens wearers in Sudan. Therefore, to calculate the sample size required, the adult population data was used. With an estimated population size of 4,000,000 adults above the age of 17 years residing in Khartoum [28]; a sample size of 405 was deemed to be representative of the adult population using GRANMO version 7.12 [29], assuming a 95% confidence interval level, a ± 5% margin of error and a 5% replacement rate.

The recommendations from the Strengthening of Reporting in Observational studies in Epidemiology (STROBE) statement have been followed in reporting the methods and results of the study (S2 Table) [30].

### Data collection tools

A questionnaire was designed by the investigators following a review of literature related to factors associated with contact lens compliance in adults [17,18,24,25]. The questionnaire (S1 Table) included questions related to the following areas:

- Participants' demographics including age, gender, occupation, level of education, type of refractive error, and smoking habits. Response options included yes/no, and multiple-choice responses.

- CL wear profile including CL power and type, CL modality and replacement schedule, source of CLs and CL products purchase. Response options included multiple-choice responses. Photographs that showed the brands of contact lenses and contact lens care products were shown to participants when they were not able to recall the product in question, a strategy that has previously been used [17,24].

- CL usage behavior including overnight wear, CL sharing, swimming and showering with CLs. Response options were: rarely (score 1), occasionally (score 2), frequently (score 3), and always (score 4).

- CL solution use: using enough solution in the lens case and checking the expiry date of the solution, with scores 1 to 4 assigned to responses ranging from "Always" to "Rarely", respectively. Further questions were about topping up the solution and sharing the solution bottle, with scores 1 to 4 assigned to responses ranging from "Rarely" to "Always", respectively.

- Hand and lens hygiene behaviors: handwashing before insertion and removal of CL, and following the correct steps for cleaning and storing the CL. Scores 1 to 4 were assigned to responses ranging from "Always" to "Rarely". In addition, rinsing the lenses with tap water was assessed with scores 1 to 4 assigned to responses ranging from "Rarely" to "Always".

- CL storage case use: cleaning the lens case with scores from 1 to 4 for responses ranging from "Always" to "Rarely", sharing lens case with scores 1 to 4 assigned to responses ranging from "Rarely" to "Always". Lens case replacement was assessed using a score of 1 for monthly replacement, score 2 for every 3 months, score 3 for 6 months and score 4 for frequency of more than 6 months.

### Data analysis

Data obtained from the questionnaire were entered into Microsoft Excel Spreadsheets then analyzed using the SPSS software version 25 (IBM Corporation, Armonk, NY, USA). Scores 1 and 2 obtained from the questionnaire were considered to be compliant, while scores 3 and 4 were deemed as non-compliant [12,25]. Numbers and percentages were calculated to summarize categorical and nominal data. Factors that were associated with non-compliance were determined using Mann-Whitney U-test and Kruskal-Wallis test. P value lower than 0.05 was deemed as statistically significant.

### Ethical consideration

Ethical approval (NU-IRB-16-09-08-1) was obtained from the Institutional Review Board (IRB) of Al Neelain University, Khartoum, Sudan. The study was conducted as per the tenets of the Declaration of Helsinki and all subjects signed a consent form prior to participation in the study. Participants were free to withdraw from the study at any point. Subjects did not receive any remuneration or other incentives for their participation in the study.

### Results and discussion

### Participants' demographics and refractive error profile

A total of 442 participants were recruited in this study and they all consented to participate. Of the 442, 92.3% (408) were women. The mean age of the participants was 24.57 ± 4.87 (range:

18 to 45 years). The majority of the participants (81.7%) either hold an undergraduate university degree or are currently pursuing an undergraduate level degree, and 45.9% were students. With regards to the refractive error distribution, the majority of participants (76.5%) were myopes. Table 1 describes the demographic and refractive error profile of the study's population.

## Contact lens wearer profile

The average duration of contact lens wear was 27.37 (± 27.42) months, ranging from 12 to 132 months. The majority of the participants (81.9%) wore soft spherical CLs. Table 2 describes the CL wearing information of the participants.

## Contact lens use, care, and hygiene behavior

Behaviors related to CL use, care product use, CL accessories, and hygiene habits were recorded as scores from 1 to 4 and the number of participants with each score is illustrated in Table 3. None of the participants reported that they wear their lens overnight or rinse their CLs with tap water either all the time (score 4) or frequently (score 3). The same responses were given to sharing CL cases. However, only half of the participants always or frequently cleaned their CL case. In addition, slightly more than one-third of the sample reported attending after-care visits always or frequently.

**Table 1. Demographic and refractive error characteristics of participants (n = 442).**

| Demographic factor | Number (%) |
|---|---|
| **Age** | |
| 18–25 | 328 (74.2%) |
| 26–35 | 99 (22.4%) |
| 36–45 | 15 (3.4%) |
| **Gender** | |
| Men | 34 (7.7%) |
| Women | 408 (92.3%) |
| **Education** | |
| High school | 18 (4%) |
| Undergraduate | 361 (81.7%) |
| Postgraduate | 63 (14.3%) |
| **Occupation** | |
| Students | 203 (45.9%) |
| Clerical | 45 (10.2%) |
| Skilled | 104 (23.5%) |
| Professional | 46 (10.4%) |
| Unemployed | 44 (10%) |
| **Smoking** | |
| Yes | 70 (15.8%) |
| No | 372 (84.2%) |
| **Refractive error** | |
| None | 53 (12%) |
| Myope | 338 (76.5%) |
| Hyperope | 27 (6.1% |
| Myopic astigmatism | 12 (2.7%) |
| Keratoconus | 12 (2.7%) |

**Table 2. Contact Lens (CL) wear profile (n = 442).**

| Information | Category | Number (%) |
|---|---|---|
| **CL type** | Soft spherical | 362 (81.9%) |
| | Soft toric | 9 (2%) |
| | Soft cosmetic | 53 (12%) |
| | Rigid-gas-permeable | 18 (4.1%) |
| **CL power** | Plano | 53 (12%) |
| | ≤5 Diopter | 338 (76.5%) |
| | >5 Diopter | 51 (11.5%) |
| **CL experience** | ≤ 24 months | 328 (74.2%) |
| | 25–48 month | 20 (4.5%) |
| | 49–72 month | 76 (17.2%) |
| | 73–96 month | 9 (2%) |
| | 97–120 month | 6 (1.4%) |
| | ≥121 month | 3 (0.75%) |
| **CL wearing time/day** | 1–5 hours | 158 (35.7%) |
| | 6–11 hours | 201 (45.5%) |
| | ≥ 12 hours | 83 (18.8%) |
| **CL wearing modality (recommended)** | Daily disposable | 15 (3.4%) |
| | Monthly replacement | 90 (20.4%) |
| | 3–6 months replacement | 145 (32.8%) |
| | Yearly replacement | 192 (43.4%) |
| **Overnight wearing (recommended)** | Yes | 5 (1.1%) |
| | No | 347 (78.5%) |
| | Not mentioned | 90 (20.4%) |
| **CL Prescriber** | Self-prescribed | 137 (31%) |
| | Eye care practitioner | 302 (68.3%) |
| | Family & friends' recommendation | 3 (0.7%) |
| **CL Place of purchase** | Contact lens practitioner | 407 (92.1%) |
| | Pharmacy | 27 (6.1%) |
| | Internet | 8 (1.8%) |
| **CL care system (type)** | Multipurpose solution | 388 (78.8%) |
| | Saline | 42 (9.5%) |
| | Do not use solution | 12 (2.7%) |
| **CL solution prescriber** | Eye care practitioner | 290 (65.6%) |
| | Pharmacy | 3 (0.7%) |
| | Was not prescribed | 149 (33.7%) |
| **CL case cleaning materials** | Multipurpose solution | 230 (52%) |
| | Saline | 86 (19.5%) |
| | Tap water | 114 (25.8%) |
| | Not applicable | 12 (2.75) |

Compliance scores obtained from Table 3 were converted into compliance rates for each behavior observed and is presented in Fig 1. Total compliance (100%) was observed for sharing CLs or a CL case and for rinsing CL with tap water. On the other hand, the least compliance rate was noted for attending after care visits to eye care practitioners (39.2%).

Based on the compliance rates observed in Fig 1, CL behaviors were grouped into 3 levels of compliance: high (>80% compliance rate), medium (40–80%), and low (<40%) [12,31]. A high compliance rate was observed for the following behaviors: sharing CLs with others,

**Table 3. Level of compliance to behaviors related to contact lens (CL) wear as reported by study's participants (n = 442).**

| Behavior | Score 1 Number (%) | Score 2 Number (%) | Score 3 Number (%) | Score 4 Number (%) |
|---|---|---|---|---|
| **Contact lens usage** | | | | |
| Overnight wearing | 425 (96.2%) | 14 (3.2%) | 3 (0.7%) | 0 (0%) |
| Sharing CL with others | 439 (99.3%) | 3 (0.7%) | 0 (0%) | 0 (0%) |
| Swimming with CL | 406 (91.9%) | 18 (4.1%) | 6 (1.4%) | 12 (2.7%) |
| Showering with CL | 314 (71%) | 50 (11.3%) | 30 (6.8%) | 48 (10.9%) |
| **CL solution usage** | | | | |
| Using enough solution in lens case | 382 (86.4%) | 39 (8.8%) | 6 (1.4%) | 15 (3.4%) |
| Topping up solution | 223 (50.4%) | 97 (21.9%) | 74 (16.7%) | 48 (10.9%) |
| Solution bottle sharing | 215 (48.7%) | 138 (31.2%) | 54 (12.2%) | 35 (7.9%) |
| Checking the expiry date of solution | 173 (39.2%) | 139 (31.4%) | 85 (19.2%) | 45 (10.2%) |
| **Hand/lens hygiene** | | | | |
| Hands washing before inserting CL | 351 (79.4%) | 56 (12.7%) | 23 (5.2%) | 12 (2.7%) |
| Hands washing before removal of CL | 272 (61.5%) | 81 (18.3%) | 59 (13.3%) | 30 (6.8%) |
| Rinsing lens with tap water | 430 (97.3%) | 12 (2.7%) | 0 (0%) | 0 (0%) |
| Rubbing, rinsing & soaking with prescribed solution | 362 (81.9%) | 47 (10.6%) | 18 (4.1%) | 15 (3.4%) |
| **CL storage case usage** | | | | |
| Cleaning lens case | 61 (13.8%) | 139 (31.4%) | 63 (14.3%) | 179 (40.5%) |
| Lens case replacement | 112 (25.3%) | 121 (27.4%) | 179 (44.6%) | 12 (2.7%) |
| Lens case sharing | 429 (97.1%) | 13 (2.95) | 0 (0%) | 0 (0%) |
| Attending after-care visits | 91 (20.6%) | 82 (18.6%) | 98 (22.2%) | 171 (38.7%) |

rinsing lenses with tap water, lens case sharing, CL overnight wearing, swimming with CL, using enough solution in lens case, cleaning and storing of CLs, handwashing before CL insertion and showering with CL. Moderate compliance levels were observed for sharing solution bottle, handwashing before CL removal, topping up solution, checking the expiry date of the solution, lens case replacement, and cleaning lens case. A low compliance level was observed only with attending after-care visits.

## Factors leading to non-compliant behaviors

A group of factors, mainly gender, education level, smoking, and CL wearing modalities, that lead to non-compliant behaviors at a statistically significant level are shown in Table 4.

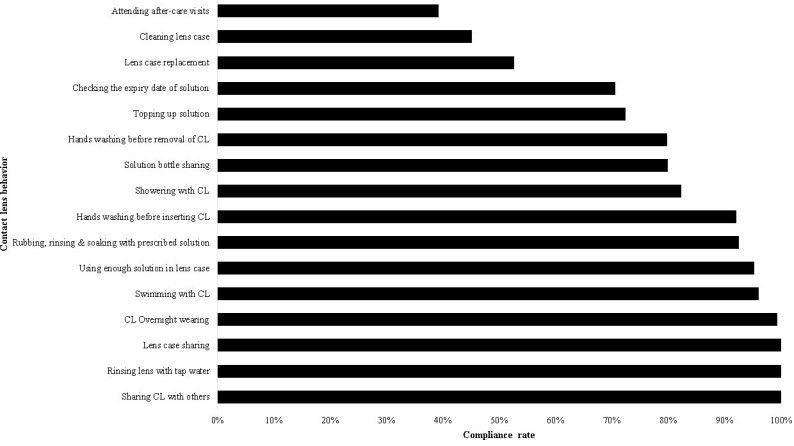

**Fig 1. Compliance rate of contact lens behaviors observed in the study sample (n = 442).**

**Table 4. Non-compliant behaviors and their associated risk factors (Mann-Whitney U-test and Kruskal-Wallis test) in Sudanese adult contact lens (CL) wearers.**

| Non-compliant Behavior | Factor leading to poor compliance | P value |
|---|---|---|
| **Contact lens usage** | | |
| Overnight wearing | High school education level | 0.006 |
| | Daily disposable CL wear | < 0.001 |
| Swimming with CL | Men | < 0.001 |
| | Self-prescribing | 0.035 |
| Showering with CL | Men | < 0.001 |
| | Daily disposable CL wear | 0.001 |
| **CL solution usage** | | |
| Using enough solution in lens case | High school education level | 0.002 |
| Topping up the solution | High school education level | 0.002 |
| | Smoking | <0.001 |
| | University students | 0.029 |
| Solution bottle sharing | Women | < 0.001 |
| | Smoking | 0.037 |
| | Cosmetic CL wear | < 0.001 |
| Checking the expiry date of the solution | High school education level | 0.001 |
| | Cosmetic CL wear | 0.014 |
| | University students | <0.001 |
| **Hand/lens hygiene** | | |
| Hands washing before inserting CL | Daily disposable CL wear | 0.001 |
| | Cosmetic CL wear | 0.018 |
| Hand washing before removal of CL | Wearing CL form more than 12 hours | 0.001 |
| | Daily disposable CL wear | 0.002 |
| Rubbing, rinsing & soaking with the prescribed solution | Smoking | 0.014 |
| | University students | 0.006 |
| | Unemployed | 0.006 |
| **CL storage case usage** | | |
| Cleaning CL case | Smoking | 0.005 |
| | University students | 0.001 |
| Lens case replacement | Smoking | < 0.001 |
| **Attending after care visits** | Daily disposable CL wear | 0.046 |
| | CL recommended by family and friends | <0.001 |
| | Cosmetic CL wear | < 0.001 |

The aim of the study was to investigate the demographics of CL wearers, the procurement and usage habits in addition to assessing the level of compliance to CL wear and care among a group of Sudanese adults. While the profile of CL wearers in many countries is well investigated [1,8,23,32], very few data exist on CL wear and care, including levels of compliance, from the context of developing countries in Africa [6,26]. The study surveyed established CL wearers residing in the Khartoum State, the political and financial center of Sudan, in addition to having the largest population compared to other states in the country [28]. Khartoum has a mixture of urban and rural population hailing from all the country's regions, which would allow a sample that better represents the Sudanese population than other states [33]. Nevertheless, our approach of convenience sampling by approaching patients during selected hours of the day would not allow for a true representation of the CL wearing population in Sudan. However, due to the political unrest in Sudan, safe and secure transportation of volunteer

optometrists was of utmost importance. In addition, not including approximately 20% of the optical stores during the data collection would have introduced selection bias in terms of non-randomness in the selection of the participants.

Soft spherical CLs are the predominant type of CL used by Sudanese adults for correction of myopia with the rest of CLs used for cosmetic reasons. This is in agreement with international trends of contact lens prescribing [4], and aligns with other reports of CL wearers profiles from Africa [6,26]. Also in alignment with other studies [5,6,12,26], was that women constituted the majority of the study population (92.3%), with many wearing CLs for cosmetic purposes. Cosmesis is one of the main reasons for women to wear CCL which is common in many Middle Eastern and African countries [5–8,26].

RGP CLs were worn by a minority of participants and were mainly for the therapeutic use of optical correction of keratoconus as observed in other countries [6,34]. Several factors can be the reason for the low number of RGP CL wearers. For example, the lack of awareness about the usefulness of RGP lenses for optically correcting keratoconus, as well as the financial burden caused by the high cost of RGPs compared to soft CLs. In addition, it has been noted that the word for RGP lenses in Arabic (the official language of Sudan), corresponds with the notion of hardness which would instill fear, as it implies inserting a 'hard' object in the eye [34]. Practitioners need to develop strategies to simplify the terminology used when prescribing RGPs to CL users and provide them with information that is context specific regarding their therapeutic benefits.

Very few toric CLs (2%) wearers were reported by the study sample, with no multifocal CLs or any specialty CLs, a trend observed in price-conscious emerging markets [5,6]. The lack of multifocal CL availability partially explains that none of the participants was above the age of 45 years. Nevertheless, monovision is an option that can be prescribed to patients; thus, CL practitioners should be aware of alternative correction strategies and communicate this to the patients during the examination session.

The overwhelming majority of participants (92.1%) purchased their lenses from a CL practitioner, whilst the rest purchased them from pharmacies or over the internet. With the availability of qualified CL practitioners in the country, it is not surprising that CL wearers choose to purchase their lenses from their practitioner [6]. This is in contrast with the observed procurement habits in the Middle East where soft CLs and cosmetic CLs can be purchased from supermarkets and beauty salons [7].

Only two-thirds of the participants were prescribed CLs or CL care products by a CL practitioner with the remaining obtaining it on their own, or they were recommended a product by a relative or a friend. In the United Kingdom, eye care practitioners are legally obliged to prescribe CLs in addition to educate the CL wearers on the proper use and care of CL [11]. However, it is not known if such practice is mandated by legislation in some other countries, where it has been noticed that CL practitioners are not actively educating CL wearers [6,12]. Lens care products are usually recommended, not prescribed by practitioners in the United States [35], thus it is important that CL practitioners in Sudan and other countries promote the use of appropriate CL solution and encourage patients to adhere with a supply of solutions consistent with the period of use of the CLs.

Several activities related to the use of water were investigated in the study with a 100% compliance rate observed for not using tap water to rinse the lenses and high compliance levels for swimming or showering while wearing CLs and for handwashing before insertion of CLs. However, a moderate level of compliance (79.8%) was observed for handwashing before CL removal. Sudan is a country that has been struggling with trachoma. Trachoma is a disease associated with the lack of hygiene, and there have been several health education campaigns for children implemented over the years to reduce the burden of trachoma. They have focused

on the risks of contaminated water and hand and face hygiene practices [36]. Perhaps the hygiene campaigns that have been targeting schoolchildren have cultivated a culture of safe water practice which would partly justify the high compliance rate. The majority of participants were university students (45.9%) attending the CL clinic at the university hospital, where proper CL handling procedures are communicated extensively both verbally and in written format. It has been reported that CL wearers who received written and verbal instructions exhibited higher levels of compliance in comparison to those who only received verbal instructions [12,17,37].

Moderate levels of compliance were noted for cleaning the CL case and frequent replacement of CL case. In addition, not checking the expiry of the solution, and not using fresh solution whenever the CL is cleaned (i.e., topping up), are behaviors that received a moderate level of compliance. Most CL cases investigated in previous studies have shown to be contaminated, especially cases that have been used for longer periods of time [19,32]. Infrequent replacement of CL case in addition to failure to clean CL cases are behaviors that should be addressed by CL practitioners in Sudan, especially as it contradicts with high levels of compliance observed in other wear and care areas. Multipurpose solution users are known for non-compliance to the above-mentioned behaviors, whereas users of hydrogen peroxide ($H_2O_2$) based solution exhibit higher compliance rates in all the aspects [35,38]. Use of $H_2O_2$ solution requires fresh solution every time the CL needs to be cleaned, while the vortex effect of the bubbles released while neutralizing the $H_2O_2$ in the lens case would replace the need for cleaning the lens case. In addition, a new case should be used with every new bottle of $H_2O_2$ solution [35]. As $H_2O_2$ was not used by any of the study's participants, due to the lack of access to $H_2O_2$ products in Sudan, CL care products importers are encouraged to introduce $H_2O_2$-based CL solutions which would encourage compliance among users.

The least level of compliance (39.2%) was observed for attendance to aftercare visits, an issue that has been reported earlier in other studies [12,25,26]. Aftercare visits are important as they allow the practitioner to objectively assess the ocular health and of the eye and the physical state of the CL. In addition, the practitioner can detect areas of non-compliance through probing questions which would allow the practitioner to then direct the appropriate message needed for the patient to continue to safely wear their CLs. CCL wearers in this study were found to be non-compliant in terms of attending aftercare visits, sharing the solution bottle, and not checking the expiry date of the solution, in addition to not washing their hands before inserting the CL. Poor hygiene and non-adherence to aftercare appointments have been noted among other behaviors among CCL wearers [3,8,19]. CCLs are considered to be safe if used as per the instructions of the CL practitioner [2]. In the United States, CCLs are considered medical devices, similar to other CLs and care products and are regulated by the Food and Drug Administration and should be fitted and prescribed by a licensed eye care practitioner [10]. However, CCLs can be perceived by the public as a cosmetic product, as in many countries they are available in pharmacies and beauty salons [7]. Thus, it is imperative for the appropriate authorities to regulate the sale of CCL and for CL practitioners to actively provide an eye health evaluation prior to prescribing CCLs in addition to actively following up on aftercare visits to gauge the level of compliance [2,3,7,8].

In the current study, DD CL wearers were observed not to comply with handwashing before lens insertion or removal, occasional overnight wearing or napping while CLs are still on, in addition not to adhere with aftercare appointments. DD CL wear is well documented to be a safe CL wear modality with a reduced risk of corneal infiltrative events in comparison to frequent replacement CLs [39]. As DD CLs are not intended for overnight wear, there is an increased chance of developing microbial keratitis while wearing DD CLs overnight [20]. In addition, handwashing before insertion or removal are required steps for both DD CLs and

reusable CLs as a method of reducing contamination of CL with fingers debris [20]. Perhaps due to the "disposable" nature of DD CLs, wearers would perceive that hand hygiene is not paramount compared to reusable CLs. In addition, as few care steps are needed for DD CL compared to reusable CLs, especially with the lack of need to purchase CL care products [20], patients would assume that aftercare visits are not essential. Thus, it is of importance that CL practitioners communicate the need to adhere to hand hygiene protocols when handling the CL and actively follow up with their patients.

University students in this study were more at risk of having non-compliant behaviors especially those related to steps required to clean the CL and cleaning the CL case. Young adult CL wearers, such as university students have been shown to be less compliant in comparison to older adults, which could be related to their imprudent lifestyle and living conditions related to staying in college halls [24,40,41]. Thus, an awareness campaign should be conducted on campus and in halls to highlight the need to comply with CL wear and care instructions.

Wearing CLs for more than 12 hours was associated with not adhering to handwashing before CL removal. It was noted that prolonged CL wear is associated with few non-compliant behaviors such as occasional napping and the use of tap water [12]. Though these specific behaviors were noted in our sample, forgetting or ignoring to wash hands before CL removal is one step that could be perceived to save time and energy, as the feeling of tiredness towards the end of the day after wearing CLs for a prolonged period of time could encourage such behavior [12]. It is therefore recommended that CL practitioners be more attentive to behaviors of CL wearers who wear their lenses for a prolonged period of time and reinforce the need to wash hands before CL removal in addition to the rest of CL wear and care procedures.

## Conclusions

The majority of CL wearers in the current study were women, under the age of 45 years and were wearing frequent replacement soft CL for the correction of myopia and using an MPS solution as the CL care product. Most of the participants were compliant with the majority of the areas investigated except for the lens case cleaning and replacement and attending aftercare visits which raises the question of the selectivity of participants to comply with instructions in certain steps and not adhering to the practitioner's advice in others. However, as smokers and students have been shown to be more at risk of not complying to lens case replacement and cleaning; it is imperative that practitioners in Sudan invest more time and provide detailed written and verbal instructions to these groups of CL wearers. DD CL wearers in addition to CCL wearers were identified to be at risk of not adhering to aftercare visits, as such, CL practitioners need to be more cautious with these groups and actively advise on the need to follow-up to avoid any complications.

## Limitations and recommendations

Similar to other studies, the assessment of compliance among CL wearers in Sudan was based on responses from the participants in a clinical setting, which has been stipulated as a source of over-estimation of the level of compliance, due to the expected reluctance of participants to provide candid answers in the clinic where they receive service [12,19]. Different results could be obtained if participants were recruited from a public setting, rather than a clinical setting, as participants' responses would be more sincere [23]. Nevertheless, the study employed a strategy where an optometrist, who was not the participants' CL practitioner, interviewed the participants to avoid such issue as previously reported in the literature [21]. In addition, the majority of contact lens wearers in the study were women, though this was expected as the majority of patients reporting to contact lens clinics are women, based on the experience of

the authors. However, a selection bias and non-randomness of the sample would have been introduced as the data was collected between the hours of 9 am and 4 pm which would exclude men whose jobs require them to work during these hours which would prevent them from accessing the clinics during these hours. Future studies should be conducted with random sampling of the whole population of Sudanese CL wearers to test the hypothesis that women indeed constitute the majority of CL wearers in Sudan.

The study did not investigate how practitioners deal with non-compliance or how they communicate with their patients regarding aspects of CL compliance, in addition, it is not known how CL wearers perceive the recommendations received by their practitioners, thus a survey investigating communication about compliance from the both the practitioners' and patients' perspective is recommended [42]. As the availability of multifocal CLs and $H_2O_2$-base solutions have not been investigated, practitioners are encouraged to communicate with CL manufacturers to make these products available which would serve a larger portion of the population. In addition, it is recommended that CL practitioners actively recommend the suitable care product to reduce the reliance on non-practitioners for advice regarding the CL solutions which could not be suitable for all CL wearers. The study design attempted to minimize confounding variables using the restriction method as per the inclusion and exclusion criteria set. However, the probable lack of availability of certain contact lenses and care products would have affected purchasing habits and the self-prescription observed. Our observational study constitutes a first step toward studying Sudanese contact lens wearers, and future hypothesis-driven studies should be carried out to confirm and expand our findings.

## Supporting information

**S1 Table. Contact lens usage and procurement questionnaire.**
(DOCX)

**S2 Table. STROBE statement checklist.**
(DOC)

**S1 Raw data.**
(XLSX)

## Acknowledgments

The authors would like to thank the following optometrists who assisted in data collection: Maab Fagier Abdelkarim, Omnia Mohammed, Reem Rashad Rushdi, Saja Mahgoup Al-Hussein.

## Author Contributions

**Conceptualization:** Yazan Gammoh.

**Data curation:** Yazan Gammoh, Mustafa Abdu.

**Formal analysis:** Yazan Gammoh.

**Methodology:** Yazan Gammoh.

**Writing – original draft:** Yazan Gammoh, Mustafa Abdu.

**Writing – review & editing:** Yazan Gammoh, Mustafa Abdu.

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
