## [Decision Letter · Decision Letter 0]

2 Mar 2021

PONE-D-21-03329

Contact lens procurement and usage habits among adults in Sudan

PLOS ONE

Dear Dr. Gammoh,

Thank you for submitting your manuscript to PLOS ONE. After careful consideration, we feel that it has merit but does not fully meet PLOS ONE’s publication criteria as it currently stands. Therefore, we invite you to submit a revised version of the manuscript that addresses the points raised during the review process.

We look forward to receiving your revised manuscript.

Kind regards,

Binaya Sapkota, PharmD

Academic Editor

PLOS ONE

Additional Editor Comments:

Please assess the reporting of this observational study of clinical data using the STROBE checklist (http://www.strobe-statement.org).

Also, address the inherent limitations of observational studies including whether there are unsupported statements of causation; and whether the analysis was affected by confounding variables, a lack of generalizability, selective reporting, post hoc analyses and other.

Journal Requirements:

2. "Please include additional information regarding the survey or questionnaire used in the study and ensure that you have provided sufficient details that others could replicate the analyses. For instance, if you developed a questionnaire as part of this study and it is not under a copyright more restrictive than CC-BY, please include a copy, in both the original language and English, as Supporting Information.

3. Please change "female” or "male" to "woman” or "man" as appropriate, when used as a noun (see for instance https://apastyle.apa.org/style-grammar-guidelines/bias-free-language/gender).

Reviewers' comments:

Reviewer's Responses to Questions

**Comments to the Author**

1. Is the manuscript technically sound, and do the data support the conclusions?

Reviewer #1: Yes

Reviewer #2: Yes

2. Has the statistical analysis been performed appropriately and rigorously? 

Reviewer #1: Yes

Reviewer #2: Yes

3. Have the authors made all data underlying the findings in their manuscript fully available?

Reviewer #1: Yes

Reviewer #2: Yes

4. Is the manuscript presented in an intelligible fashion and written in standard English?

Reviewer #1: Yes

Reviewer #2: No

5. Review Comments to the Author

Reviewer #1: It is a nicely written paper. The paper looks into the aspects of contact lens wear and care habits among wearers in Sudan. Results show a relatively high to moderate levels of compliance to contact lenses. In overall, the introduction reads well and sets the tone for the aim of the study. Methods are described in detail and results are appropriately summarised. Discussion is slightly long but discusses all the relevant points. There are few minor punctuation errors. My specific comments are attached below:

Line 100, 107 and 119: Replace semicolon with a comma

Line 122: delete one ‘in’

Methods: How were the participants selected, was it random or everyone attending the clinics and hospitals? There might be a selection bias, which needs to be clarified, especially because around 92% participants were female. Was it because of selection method or is it a true representative of the population?

Line 222: Table 2- percent in CL power column doesn’t add to 100.

Line 230: clean should be ‘cleaned’

Line 253: Please mention what statistical tests were used to derive these p values.

Line 398: The statement ‘The majority of CL wearers in Sudan are females’ is not a correct statement. Authors did not collect the data to check this – this is not a population-based study but a cross-sectional study of a sample. Instead, they can write that a majority of CL wearers were female in the present study (not in Sudan).

Reviewer #2: manuscript is not presented in an intelligible fashion and written in standard English.

Deeper analysis can be done

Please make sure that you have followed the authors guidelines properly.

Please keep limitations and recommendation sections after conclusion section

6. PLOS authors have the option to publish the peer review history of their article (what does this mean?). If published, this will include your full peer review and any attached files.

Reviewer #1: No

Reviewer #2: **Yes: **Sunil Shrestha

---

## [Author Response · Author response to Decision Letter 0]

11 Mar 2021

Dear Binaya Sapkota, PharmD

I would like to thank you and the esteemed reviewers for all the valuable comments and suggestions. Kindly find below a reply to the reviewers’ and editor’s comments. I sincerely hope it meets your requirements.

1. Additional editor’s comments:

Comment: Please assess the reporting of this observational study of clinical data using the STROBE checklist

Response: Thank you for raising this important point. A sentence about STROBE statement has been included in the methods section Lines 165-167

Supplement S1 Table includes the STROBE checklist.

Comment: Also, address the inherent limitations of observational studies including whether there are unsupported statements of causation; and whether the analysis was affected by confounding variables, a lack of generalizability, selective reporting, post hoc analyses and other.

Response: I would like to thank you for binging my attention to this issue. The requested information has been included in the limitations section. Lines 431-442

2. Reviewer 1 comments:

Comment: Line 100, 107 and 119: Replace semicolon with a comma

Response: Thank you so much for raising my attention to this issue. The semicolon in each line has been changed to a comma.

Comment: Line 122: delete one ‘in’

Response: One ‘in’ has been deleted.

Comment: Methods: How were the participants selected, was it random or everyone attending the clinics and hospitals? There might be a selection bias, which needs to be clarified, especially because around 92% participants were female. Was it because of selection method or is it a true representative of the population?

Response: The following paragraph was added to the methods section: ‘All patients attending the clinics during the data collection period were invited to participate in the study. To further minimize selection bias, and to ensure full participation of patients, volunteer optometrists administered the questionnaire rather than the patients’ contact lens practitioners.’

The following statement was added in the limitations section: ‘In addition, the majority of contact lens wearers in the study were women, though this was expected as the majority of patients reporting to contact lens clinics are women, based on the experience of the authors.’

Comment: Line 222: Table 2- percent in CL power column doesn’t add to 100.

Response: The percent has been edited and now add to 100

Comment: Line 230: clean should be ‘cleaned’

Response: Clean has been changed to “cleaned”

Comment: Line 253: Please mention what statistical tests were used to derive these p values.

Response: Mann-Whitney U-test and Kruskal-Wallis test were mentioned in table 4 title.

Comment: Line 398: The statement ‘The majority of CL wearers in Sudan are females’ is not a correct statement. Authors did not collect the data to check this – this is not a population-based study but a cross-sectional study of a sample. Instead, they can write that a majority of CL wearers were female in the present study (not in Sudan).

Response: The following statement was used instead in conclusion section line 412: 

‘The majority of CL wearers in the current study were women’

3. Reviewer's 2 comments:

Comment: manuscript is not presented in an intelligible fashion and written in standard English.

Response: Sections have been arranged as per the style requirement of the journal.

Subsections in what was previously the discussion section were removed, as the results and discussion section were merged as per the Journal’s style requirements.

Reporting of males and females as men and women when used as nouns have been corrected.

Spelling and grammar has been checked throughout the manuscript.

Comment: Deeper analysis can be done

Response: STROBE statement checklist has been added.

Inherent limitations to observational studies have been addressed.

Type of statistical test used in table 4 was included.

Comment: Please make sure that you have followed the authors guidelines properly.

Response: Authors’ guidelines and PLOS ONE’s style requirements have been followed.

Comment: Please keep limitations and recommendation sections after conclusion section

Response: Limitations and recommendations section is placed after the conclusion section, Lines 430-451

4. Journal's requirements

Comment: Please ensure that your manuscript meets PLOS ONE's style requirements, including those for file naming.

Response: Manuscript, tables, figures, and supplement files have been edited to meet PLOS ONE’s style requirements.

Comment: Please include additional information regarding the survey or questionnaire used in the study and ensure that you have provided sufficient details that others could replicate the analyses.

Response: The questionnaire has been included as supplement S2 Table.

Comment: Please change "female” or "male" to "woman” or "man" as appropriate, when used as a noun

Response: All changes needed have been implemented

Comment: Your ethics statement should only appear in the Methods section of your manuscript.

Response: The ethics statement has been only included in the methods section.

I would like to inform you that the changes that have been made based on the reviewer’s comments have enriched the manuscript, thank you for your consideration.

Yours sincerely,

Yazan Gammoh, PhD

---

## [Editor Report · Decision Letter 1]

24 Mar 2021

PONE-D-21-03329R1

Contact lens procurement and usage habits among adults in Sudan

PLOS ONE

Dear Dr. Gammoh,

Thank you for submitting your manuscript to PLOS ONE. After careful consideration, we feel that it has merit but does not fully meet PLOS ONE’s publication criteria as it currently stands. Therefore, we invite you to submit a revised version of the manuscript that addresses the points raised during the review process.

We look forward to receiving your revised manuscript.

Kind regards,

Binaya Sapkota, PharmD

Academic Editor

PLOS ONE

Additional Editor Comments (if provided):

Thank you very much for submitting your paper to PLOS ONE. While reviewing the 'Responses to the Editor and Reviewers', it seems that you have not addressed following key concerns:

1. Editor's concern: Comment related to the inherent limitations of observational studies; effect of the confounding variables, a lack of generalizability, selective reporting, post hoc analyses and other.

2. Reviewer 1 concern: Comments related to selection of the participants (random or non-random), selection bias, true representativeness of the population.

These comments should be addressed before it can be further considered by the external reviewers.

---

## [Author Response · Author response to Decision Letter 1]

13 Apr 2021

Dear Binaya Sapkota, PharmD

I would like to thank you and the esteemed reviewers for all the valuable comments and suggestions. Kindly find below a reply to the reviewers’ and editor’s comments. I sincerely hope it meets your requirements.

1. Additional editor’s comments:

Comment: Please assess the reporting of this observational study of clinical data using the STROBE checklist

Response: Thank you for raising this important point. A sentence about STROBE statement has been included in the methods section. Lines 170-172

Supplement S2 Table includes the STROBE checklist.

Comment: Also, address the inherent limitations of observational studies including whether there are unsupported statements of causation; and whether the analysis was affected by confounding variables, a lack of generalizability, selective reporting, post hoc analyses and other.

Response: I would like to thank you for bringing my attention to this issue. The requested information has been included in the methods section (Lines 154-162) and limitations section (Lines 428-433, Lines 443-448). No post-hoc analyses have been done. No selective reporting has been practiced. 

2. Reply to Reviewer 1’s comments:

Comment: Line 100, 107 and 119: Replace semicolon with a comma

Response: Thank you so much for raising my attention to this issue. The semicolon in each line has been changed to a comma. Lines 98, 105, 117

Comment: Line 122: delete one ‘in’

Response: One ‘in’ has been deleted. Line 120

Comment: Methods: How were the participants selected, was it random or everyone attending the clinics and hospitals? There might be a selection bias, which needs to be clarified, especially because around 92% participants were female. Was it because of selection method or is it a true representative of the population?

Response: The following paragraph was added to the methods section, Lines 154-157: ‘All patients attending the clinics during the data collection period were invited to participate in the study. To further minimize selection bias, and to ensure full participation of patients, volunteer optometrists administered the questionnaire rather than the patients’ contact lens practitioners.’

To further clarify the patient’s selection process the following paragraph was added to the methods section, Lines 158-162 : ‘Clinics in Khartoum operate from 9 am to 7 pm, however, volunteer optometrists approached all the patients attending the contact lens clinics between 10 am and 4 pm to allow for safe and secure transportation of volunteers. The same convenient sampling hours were also adopted for optical stores which usually operate till 9 pm. Approximately 20% of the optical stores in Khartoum were not surveyed due to difficulty in securing transportation for the volunteers.’

The following statement was added in the limitations section, Lines 425-433: ‘In addition, the majority of contact lens wearers in the study were women, though this was expected as the majority of patients reporting to contact lens clinics are women, based on the experience of the authors. However, a selection bias and non-randomness of the sample would have been introduced as the data was collected between the hours of 9 am and 4 pm which would exclude men whose jobs require them work during these hours which would prevent them from accessing the clinics during these hours. Future studies should be conducted with random sampling of the whole population of Sudanese CL wearers to test the hypothesis that women indeed constitute the majority of CL wearers in Sudan.’

Comment: Line 222: Table 2- percent in CL power column doesn’t add to 100.

Response: The percent has been edited and now add to 100

Comment: Line 230: clean should be ‘cleaned’

Response: Clean has been changed to “cleaned”, Line 246

Comment: Line 253: Please mention what statistical tests were used to derive these p values.

Response: Mann-Whitney U-test and Kruskal-Wallis test were mentioned in table 4 title.

Comment: Line 398: The statement ‘The majority of CL wearers in Sudan are females’ is not a correct statement. Authors did not collect the data to check this – this is not a population-based study but a cross-sectional study of a sample. Instead, they can write that a majority of CL wearers were female in the present study (not in Sudan).

Response: The following statement was used instead: 

‘The majority of CL wearers in the current study were women’, Line 404

3. Reply to Reviewer 2’s comments:

Comment: manuscript is not presented in an intelligible fashion and written in standard English.

Response: Sections have been arranged as per the style requirement of the journal.

Subsections in what was previously the discussion section were removed, as the results and discussion section were merged as per the Journal’s style requirements.

Reporting of males and females as men and women when used as nouns have been corrected.

Spelling and grammar have been checked throughout the manuscript.

Comment: Deeper analysis can be done

Response: STROBE statement checklist has been added.

Inherent limitations to observational studies have been addressed.

Type of statistical test used in table 4 was included.

Comment: Please make sure that you have followed the authors guidelines properly.

Response: Authors’ guidelines and PLOS ONE’s style requirements have been followed. All manuscript and files

Comment: Please keep limitations and recommendation sections after conclusion section

Response: Limitations and recommendations section is placed after the conclusion section. Lines 417-448

Requirement 1: Please ensure that your manuscript meets PLOS ONE's style requirements, including those for file naming.

Response: Manuscript, tables, figures, and supplement files have been edited to meet PLOS ONE’s style requirements.

Requirement 2: Please include additional information regarding the survey or questionnaire used in the study and ensure that you have provided sufficient details that others could replicate the analyses.

Response: The questionnaire has been included as supplement S1 Table.

Requirement 3: Please change "female” or "male" to "woman” or "man" as appropriate, when used as a noun

Response: All changes needed have been implemented

Requirement 4: Your ethics statement should only appear in the Methods section of your manuscript.

Response: The ethics statement has been only included in the methods section.

---

## [Decision Letter · Decision Letter 2]

20 Apr 2021

PONE-D-21-03329R2

Contact lens procurement and usage habits among adults in Sudan

PLOS ONE

Dear Dr. Gammoh,

Thank you for submitting your manuscript to PLOS ONE. After careful consideration, we feel that it has merit but does not fully meet PLOS ONE’s publication criteria as it currently stands. Therefore, we invite you to submit a revised version of the manuscript that addresses the points raised during the review process.

ACADEMIC EDITOR: Thank you very much for submitting your paper to PLOS ONE. After careful editorial consideration and evaluation of the reviewers' reports, the authors are suggested to go through the English language editing before it can be accepted for publication.

We look forward to receiving your revised manuscript.

Kind regards,

Binaya Sapkota, PharmD

Academic Editor

PLOS ONE

Journal Requirements:

Reviewers' comments:

Reviewer's Responses to Questions

**Comments to the Author**

1. If the authors have adequately addressed your comments raised in a previous round of review and you feel that this manuscript is now acceptable for publication, you may indicate that here to bypass the “Comments to the Author” section, enter your conflict of interest statement in the “Confidential to Editor” section, and submit your "Accept" recommendation.

Reviewer #1: All comments have been addressed

Reviewer #2: All comments have been addressed

2. Is the manuscript technically sound, and do the data support the conclusions?

Reviewer #1: Yes

Reviewer #2: Partly

3. Has the statistical analysis been performed appropriately and rigorously? 

Reviewer #1: Yes

Reviewer #2: Yes

4. Have the authors made all data underlying the findings in their manuscript fully available?

Reviewer #1: Yes

Reviewer #2: Yes

5. Is the manuscript presented in an intelligible fashion and written in standard English?

Reviewer #1: Yes

Reviewer #2: No

6. Review Comments to the Author

Reviewer #1: I congratulate the authors for their comprehensive work on this manuscript. I believe they have addressed all my comments. These types of studies are necessary to educate people because contact lens care behaviour can be different throughout the world. One of the major concerns with contact lens usage is the incidence of microbial keratitis, which is due to poor hygiene behaviour. While there may be tendency to report data from places where there is relatively an easy access to research funds, the literature needs data from low income settings as well, so that we can make valid comparisons. Authors have outlined their limitations and have addressed my concerns. There are few issues with the use of punctuation, especially the use of semicolon, which could be because of English as a second language.

Reviewer #2: The authors have addressed most of the comments from the reviewers. However, an English editing is needed.

7. PLOS authors have the option to publish the peer review history of their article (what does this mean?). If published, this will include your full peer review and any attached files.

Reviewer #1: **Yes: **Sanjay Marasini

Reviewer #2: **Yes: **Sunil Shrestha

---

## [Author Response · Author response to Decision Letter 2]

30 Apr 2021

Re: PONE-D-21-03329

Contact lens procurement and usage habits among adults in Sudan

30 April 2021

Dear Binaya Sapkota, PharmD

I would like to thank you and the esteemed reviewers for all the valuable comments and suggestions. Allow me to thank Reviewer 1 for their encouraging words and the support expressed to publish data from developing countries with low-income like Sudan. 

1. As both reviewers highlighted the need for English language editing, all the necessary changes are shown in the file “Revised Manuscript with Track Changes” to allow the esteemed reviewers to track all the changes. All the tracked changes have been incorporated into the manuscript text in the file “Manuscript”. 

2. As the reviewers had specific comments regarding the inappropriate use of commas and semicolons, kindly find below the changes made specifically for the use of semicolons:

Line 47 

original: lenscase; 

changes: lenscase,

Line 56 

original: appointments; 

changes: appointments,

Line 83 

original: worldwide; 

changes: worldwide,

Line 98 

original: beauty salons and over the internet 

changes: beauty salons, and over the internet

Line 122 

original: infrequent cleaning and discarding of the lens case; improper use of lens care products; use of tap changes: infrequent cleaning and discarding of the lens case, improper use of lens care products, use of tap

Line 125 

original: compliance to CL wear and care; age, gender and level of education are risk factors 

changes: compliance to CL wear and care, age, gender, and level of education are risk factors

Line 132

original: CL wearers such as daily disposable (DD) CL wearers [21]; females [7]; or students 

changes: CL wearers such as daily disposable (DD) CL wearers [21], females [7], or students

Line 233 

original: Adult CL wearers; 18 years of age and above, 

changes: Adult CL wearers 18 years of age and above,

Line 269 

original: Participants’ demographics including, age; gender; occupation; level of education, type of changes: Participants’ demographics including age, gender, occupation, level of education, type of

Line272 

original: CL wear profile including CL power and type; CL modality and replacement schedule; source 

changes: CL wear profile including CL power and type, CL modality and replacement schedule, source

Line 275 

original: to participants when they were not able to recall the product in question; a strategy that has 

changes: to participants when they were not able to recall the product in question, a strategy that has

Line 292 

original: CL usage behavior including overnight wear; CL sharing; swimming and showering 

changes: CL usage behavior including overnight wear, CL sharing, swimming and showering

Line 296 

original: with scores 1 to 4 assigned to responses ranging from “Always” to “Rarely”; 

changes: with scores 1 to 4 assigned to responses ranging from “Always” to “Rarely”,

Line 298 

original: ranging from “Rarely” to “Always”; respectively. 

changes: ranging from “Rarely” to “Always”, respectively.

Line 299 

original: before insertion and removal of CL; and 

changes: before insertion and removal of CL, and

Line 304 

original: “Always” to “Rarely”; sharing lens case 

changes: “Always” to “Rarely”, sharing lens case

Line 351-2 

original: (range; 18 to 45 years). 

changes: (range: 18 to 45 years).

Line 400 

original: Based on the compliance rates observed in Fig1; contact lens behaviors 

changes: Based on the compliance rates observed in Fig1, CL behaviors

Lines 405-7 

original: showering with CL. Moderate compliance level observed for sharing solution bottle; handwashing before CL removal; topping up solution; checking the expiry date of the solution; lens case replacement and cleaning lens case. 

changes: showering with CL. Moderate compliance levels were observed for sharing solution bottle, handwashing before CL removal, topping up solution, checking the expiry date of the solution, lens case replacement, and cleaning lens case.

Line 410 

original: A group of factors; mainly gender, 

changes: A group of factors, mainly gender,

Line 432 

original: CL wear and care; including levels of 

changes: CL wear and care, including levels of

Line 512 

original: struggling with trachoma; a disease 

changes: struggling with trachoma. Trachoma is a disease

Line 518 

original: at the university hospital; where proper CL 

changes: at the university hospital, where proper CL

Line 551 

original: any of the study’s participants; due to the lack of access to 

changes: any of the study’s participants, due to the lack of access to

Line 582 

original: beauty salons [7]; thus, it is 

changes: beauty salons [7]. Thus, it is

Line 646 

original: of the level of compliance; due to the expected reluctance 

changes: of the level of compliance, due to the expected reluctance

Yours sincerely,

Yazan Gammoh, PhD

---

## [Editor Report · Decision Letter 3]

7 May 2021

Contact lens procurement and usage habits among adults in Sudan

PONE-D-21-03329R3

Dear Dr. Gammoh,

We’re pleased to inform you that your manuscript has been judged scientifically suitable for publication and will be formally accepted for publication once it meets all outstanding technical requirements.

Kind regards,

Binaya Sapkota, PharmD

Academic Editor

PLOS ONE
---

## [Editor Report · Acceptance letter]

11 May 2021

PONE-D-21-03329R3 

Contact lens procurement and usage habits among adults in Sudan 

Dear Dr. Gammoh:

I'm pleased to inform you that your manuscript has been deemed suitable for publication in PLOS ONE. Congratulations! Your manuscript is now with our production department. 

Kind regards, 

on behalf of

Dr. Binaya Sapkota 

Academic Editor

PLOS ONE